# Quality of Automated Stereotactic Radiosurgery Plans in Patients with 4 to 10 Brain Metastases

**DOI:** 10.3390/cancers13143458

**Published:** 2021-07-10

**Authors:** Anna Petoukhova, Roland Snijder, Rudolf Wiggenraad, Linda de Boer-de Wit, Ivonne Mudde-van der Wouden, Mireille Florijn, Jaap Zindler

**Affiliations:** 1Department of Medical Physics, Haaglanden Medical Center, 2262 BA Leidschendam, The Netherlands; r.snijder@haaglandenmc.nl (R.S.); mireille.florijn@emailmij.org (M.F.); 2Department of Radiation Oncology, Haaglanden Medical Center, 2262 BA Leidschendam, The Netherlands; r.wiggenraad@gmail.com (R.W.); ledbdw@gmail.com (L.d.B.-d.W.); i.muddevan.der.wouden@haaglandenmc.nl (I.M.-v.d.W.); or j.zindler@hollandptc.nl (J.Z.); 3Holland Proton Therapy Centre, 2629 JH Delft, The Netherlands

**Keywords:** multiple brain metastases, linac-based stereotactic radiosurgery or hypofractionated stereotactic radiotherapy, automated planning, single isocenter, IMRT, DCA, VMAT

## Abstract

**Simple Summary:**

Stereotactic radiosurgery (SRS) and hypofractionated stereotactic radiotherapy (SRT) are promising treatment options for patients with multiple brain metastases in the current era of personalized medicine. Recent international guidelines propose SRS also in patients with more than three brain metastases with low-volume disease. Optimal treatment quality with sparing of healthy brain tissue is essential to avoid SRS/SRT complications such as brain necrosis. The aim of this study was to compare linac (linear accelerator)-based SRS/SRT plan quality of automated planning, intensity modulated radiotherapy (IMRT), volumetric modulated arc radiotherapy (VMAT) and manually planned dynamic conformal arc (DCA) plans as well as single- and multiple-isocenter techniques. We found that automated planning with DCA or IMRT can make linac-based SRS/SRT plan quality with single isocenter comparable with a manually planned DCA plan with a separate isocenter for each metastasis.

**Abstract:**

The purpose was to compare linac-based stereotactic radiosurgery and hypofractionated radiotherapy plan quality of automated planning, intensity modulated radiotherapy (IMRT) and manual dynamic conformal arc (DCA) plans as well as single- and multiple-isocenter techniques for multiple brain metastases (BM). For twelve patients with four to ten BM, seven non-coplanar linac-based plans were created: a manually planned DCA plan with a separate isocenter for each metastasis, a single-isocenter dynamic IMRT plan, an automatically generated single-isocenter volumetric modulated arc radiotherapy (VMAT) plan, four automatically generated single-isocenter DCA plans with three or five couch angles, with high or low sparing of normal tissue. Paddick conformity index, gradient index (GI), mean dose, total V_12Gy_ and V_5Gy_ of uninvolved brain, number of monitor units (MUs), irradiation time and pass rate were compared. The GI was significantly higher for VMAT than for separate-isocenter, IMRT, and all automatically generated plans. The number of MUs was lowest for VMAT, followed by automatically generated DCA and IMRT plans and highest for manual DCA plans. Irradiation time was the shortest for automatically planned DCA plans. Automatically generated linac-based single-isocenter plans for multiple BM reduce the number of MUs and irradiation time with at least comparable GI and V_5Gy_ relative to the reference separate-isocenter DCA plans.

## 1. Introduction

Whole brain radiotherapy (WBRT) has been the standard treatment of multiple brain metastases (BM) for many years. WBRT has side effects such as fatigue, hair loss, and neurocognitive damage with only a low local tumor control probability. Stereotactic radiosurgery (SRS) has overcome several of these limitations [1]. According to the guidelines of the American Society for Radiation Oncology, SRS alone, WBRT and SRS, or WBRT alone should be considered for selected patients with multiple BM. These guidelines only support SRS without concurrent WBRT for patients with up to four BM [2].

In 2014, Yamamoto et al. [3] published a multi-institutional prospective observational trial (JLGK0901) of patients with one to ten newly diagnosed brain metastases (largest tumor <10 cm^3^ in volume and <3 cm in longest diameter; total cumulative volume ≤15 cm^3^). The authors concluded that SRS without WBRT as the initial treatment for patients with five to ten BM is non-inferior to SRS alone in those with two to four brain metastases, in terms of overall survival [3]. This study opened the discussion as to whether SRS is an alternative treatment option for patients with more than three BM with low-volume disease. SRS is a promising treatment option for patients with four or more BM and randomized trials are ongoing to determine its value [4]. According to UK NICE guidelines, SRS can be considered for people with multiple BM who have controlled or controllable extracranial disease and Karnofsky performance status of 70 or higher [5].

In the radiotherapy literature, it is known that hypofractionated stereotactic radiotherapy (SRT) may decrease late side effects, compared to single-fraction treatment (SRS) [6]. To achieve a local control rate of more than 90% after one year, it is important to deliver a cumulative biologically effective dose (BED) of at least 50 Gy. Delivering 30 Gy in three fractions or 37.5 Gy in five fractions results in a BED of more than 50 Gy [7,8]. The volumetric study by Putz et al. provided initial evidence that the improvements in therapeutic ratio expected for SRT in larger brain metastases might equally extend into the domain of smaller metastases, traditionally less considered for fractionated treatment [9].

There are some studies that have compared SRS platforms for treating multiple BM. Ma et al. reported that the dose delivered to normal brain is dependent on the radiosurgery platform: Gamma Knife (superior), linear accelerator (linac) or CyberKnife [10]. SRS plans for the treatment of multiple metastases with single-isocenter volumetric modulated arc (VMAT) resulted in comparable dose gradients in the high dose area, but the mean brain dose with VMAT was higher than with the other techniques [11]. Based on two cases involving three and seven BM, Eaton et al. compared treatment platforms of 21 clinical centers and concluded that most platforms were able to achieve comparable plans, except for the smallest volumes and when larger planning margins are used [12].

Traditionally, linac-based SRS techniques have utilized separate-isocenter plans for the treatment of patients with multiple BM, whereby each lesion is associated with one isocenter placed in the geometric center of the lesion. Different planning techniques to irradiate multiple BM with a single isocenter can be used, such as dynamic conformal arcs (DCA), intensity modulated radiotherapy (IMRT) or VMAT. A few studies have been published that reported the irradiation of multiple BM with a single isocenter [12,13,14,15,16,17,18,19]. Most of these plans were calculated for a Varian or Novalis accelerator, whereas only a few comparisons were performed for an Elekta accelerator [17,19]. Single-isocenter linac-based SRS for multiple BM is more efficient, and thereby patient-friendly, than the Gamma Knife, because multiple BM are treated within half an hour as opposed to over an hour with the Gamma Knife. Besides the SRS Gamma Knife being more time-consuming, an invasive frame is required, making the treatment less patient-friendly as well. In a palliative setting, an efficient and patient-friendly treatment is desirable.

The purpose of this study was to investigate whether linac-based SRS/SRT plan quality of single-isocenter plans, either automatically generated (DCA and VMAT) or using dynamic (sliding window) IMRT, could be comparable or even superior to the plan quality of manually planned DCA plans with separate isocenters for each metastasis as a reference. In our study, seven out of twelve patients had a total cumulative volume >15 cm^3^, whereas the previous published studies according to Yamamoto et al. [3] included patients with smaller metastases. Moreover, we compared the dose in the critical structures (healthy brain, brain stem, chiasm, optic nerves, eyes, and pituitary gland). All plans were created for an Elekta accelerator with an Agility multi-leaf collimator.

## 2. Materials and Methods

### 2.1. Patient Selection, Imaging, and Dose Prescription

Patients with brain metastases <4 cm in diameter were referred for SRS/SRT if they had a Karnofsky performance score >60 and a prognosis of more than six months. Twelve patients with four to ten brain metastases (median = 6) with a total of 77 lesions were selected for this study. Nine of these patients were treated at the Haaglanden Medical Center (HMC). Three patients (#6, 8, and 11) were anonymized benchmark patients within a multicenter randomized phase III trial (ClinicalTrials.gov Identifier: NCT02353000) [20]. Most of the studied patients were included in this trial.

For each HMC patient, the acquisition of the planning CT was conducted on a CT scan (Brilliance Big Bore CT scanner, PHILIPS, Cleveland, OH, USA), slice thickness 1.5 mm and a pixel size of 0.7 × 0.7 mm^2^, with a thermoplastic mask (Q-fix, Avondale, AZ, USA) in combination with the Precise Bite mouth piece (CIVCO Medical Solutions, Kalona, IA, USA). For the three benchmark patients, a three-point thermoplastic mask (MarcoMedics, Moordrecht, The Netherlands) was used. In addition, all patients had an MRI of the brain with the following sequences: T1-weighted imaging without and with gadolinium (voxel size 1.1 × 1.1 × 1.3 mm^3^), T2-weighted imaging, diffusion-weighted imaging, and MR perfusion imaging, using a SE-EPI sequence. The time between the CT and MRI scans was less than one week.

The gross target volumes (GTVs) were delineated by an experienced radiation oncologist as the area of contrast enhancement on a T1-weighted MRI. This sequence was also used for auto-segmentation of organs at risk (OARs) (Anatomical Mapping Elements, Brainlab, Munich, Germany) and edited afterwards by the radiation oncologist according to the atlas for neuro-oncology [21]. The GTV to planning target volume (PTV) margin was 1 mm for all metastases. This GTV-PTV margin was used for all BM based on online Cone Beam CT position verification and a 6D Hexapod couch (Elekta, Stockholm, Sweden) correction for rotations [22,23]. Kirkpatrick et al. compared 1 and 3 mm GTV-PTV margins for BM and found no significant difference in terms of local control [24]. Dose was clinically prescribed, depending on the PTV of the largest metastasis: 1× 21 Gy for a PTV of 1–10 cm^3^, 1 × 18 Gy for a PTV of 10–20 cm^3^, and for a PTV > 20 cm^3^ three fractions of 8.5 Gy [8], and the same prescription dose was used for all other metastases according to the multicenter randomized phase III trial [20]. This choice was made because of the difficulty of radiobiological combination of a single fraction with three fractions for the same patient. The PTV coverage should be at least 99%. Table 1 shows the number of lesions, their volumes, and the prescription dose for each patient.

OAR constraints were based on dose limits for these organs in quantitative analyses of normal tissue in the clinic papers (QUANTEC papers) [25,26]. No planning organ at risk volume (PRV) was used for OARs.

### 2.2. Treatment Planning

For each patient, seven non-coplanar linac-based SRS/SRT plans were created by an experienced planner and reviewed by two experienced planners: a manually planned DCA plan with separate isocenters and 2–5 couch angles each with an arc of approximately 100° for each metastasis (iPlan RT Dose, v.4.5.4, BrainLAB, Munich, Germany), an inversely planned dynamic IMRT plan with three couch angles and ten beams (iPlan RT Dose, v.4.5.4, BrainLAB), an Auto-Planning VMAT plan with dual arcs at three couch angles (Pinnacle, v.9.10, PHILIPS Medical Systems, Cleveland, OH, USA), two automatically generated DCA plans with three or five couch angles (Multiple Brain Mets SRS Elements, v.1.6, BrainLAB), and two automatically generated DCA plans with five couch angles with high and low sparing of normal tissue (NT) of the brain (Multiple Brain Mets SRS Elements, v.2.0). Manually planned DCA plans with a separate isocenter for each metastasis were used as a reference.

The last six plans used a single isocenter (see an example in Figure 1). The Auto-Planning engine is designed in Pinnacle to create an optimized plan with minimal user interaction for IMRT and VMAT treatment planning; the Auto-Planning module is able to create treatment plans with consistent quality using a single optimization preset including beam set-up, dose prescription, objectives and priorities for OARs, and PTVs. In iPlan, ring structures were used for modulation of the dose gradient of IMRT plans. In Multiple Brain Mets SRS Elements (MBM Elements also known as automated single-isocenter DCA (SIDCA)), once the prescription dose and the dose-volume criteria to the PTVs were specified, the dose optimization was automatically performed without user interaction. The treatment plan optimization is composed of two main optimization steps: distribution of metastases to arcs and collimator optimization followed by core optimization: inverse treatment plan optimization by simultaneous optimization of beam’s eye view (BEV) margins, clippings, and arc weights [15].

A Versa HD linac (Elekta, Stockholm, Sweden) with 6 MV photon energy with flattening filter (dose rate = 500 monitor units (MUs)/min) equipped with an integrated Agility multi-leaf collimator (MLC, Elekta), consisting of 160 interdigitating leaves with a width of 5 mm, was modelled for all treatment planning purposes. Monte Carlo and Collapsed Cone algorithms were used for dose calculation with a dose grid of 1 mm in iPlan RT Dose/MBM Elements and Pinnacle, respectively.

In Pinnacle, a three dual VMAT arcs with a collimator angle of 45° arrangement spanning of 360° at couch angle of 0° and 160° at couch angle of 60° and 300° was chosen. In automated single-isocenter DCA, the default number of couch angles was five (0°, 20°, 60°, 280°, and 320°). In addition to these five angles, we also used three couch angles (0°, 60°, and 300°) in order to make a comparison with the other three-couch angle techniques and to study the influence of five instead of three couch angles on the plan quality. The algorithm of MBM Elements v.1.6 starts by considering a maximum of five couch angles and two independent arcs of 160° per couch angle, but only relevant arcs remain [13]. In v.2.0, all ten arcs were used by default and jaw tracking of the aperture of the MLC to reduce doses to normal tissues around metastases was possible. Additionally, we were able to use the “spare NT” (normal tissue) slider (position at high or low) with the aim of sparing normal brain tissue [27].

### 2.3. Plan Comparison

Separate-isocenter DCA (separate), IMRT, VMAT, MBM Elements v.1.6 with three couch angles (SIDCA_1), MBM Elements v.1.6 with five couch angles (SIDCA_2), MBM Elements v.2.0 with high NT (SIDCA_3), and low NT (SIDCA_4) plans were compared for 12 patients.

The plans were evaluated by visual inspection and by analyses of dose volume histograms (DVHs) from each TPS for the PTVs and OARs. For the critical structures (brain stem, chiasm, optic nerves, eyes, and pituitary gland) D_max_, dose to 0.03 cc, was calculated. For separate-isocenter DCA plans, a summation plan in iPlan was used.

To assess SRS/SRT plan quality, the Paddick conformity index (CI, high CI correlates with favorable plan quality) [28], the Paddick gradient index (GI, low GI correlates with favorable plan quality) [29] as primary endpoint, the mean dose, the total V_12Gy_ and V_5Gy_ of the uninvolved brain, i.e., brain volume—GTVs (low V_12Gy_ and V_5Gy_ correlate with favorable plan quality [30]), were studied and reported as mean ±1 standard deviation (SD). Additionally, the number of MUs and irradiation time were compared. Irradiation time was calculated using Mobius 3D software version 3.1 (Varian, Palo Alto, CA, USA). As quality analysis, a 3D global gamma comparison between that calculated in the treatment planning system and in Mobius absolute dose distributions over the entire dose calculation volume was performed with 3% dose difference (of the maximum dose) and 3 mm distance-to-agreement (DTA) criteria according to Van Dyk [31,32].

### 2.4. Statistics

To assess the difference in plan quality, analysis of variance (ANOVA) for repeated measures was performed for dosimetric parameters between various planning techniques. Paired T-tests were performed to compare the techniques pairwise. For statistical analysis, IBM SPSS version 22 (IBM Corp, Armonk, NY, USA) was used. A two-sided *p*-value ≤ 0.05 was considered statistically significant.

## 3. Results

The mean total CI was the highest (0.73 ± 0.1) for dynamic IMRT and automated SIDCA_4 (MBM Elements v.2.0 with low NT) (see Figure 2, Figure 3, and Table 2). According to the ANOVA results, the CI did not differ significantly across the seven planning techniques. Similar results were found for ten patients (#9 and 12 excluded) irradiated in a single fraction (see Appendix A).

The GI was significantly higher (worse) for VMAT plans (7.1 ± 3.0) compared to separate-isocenter DCA (3.7 ± 0.6, *p* = 0.003), IMRT (4.8 ± 1.1, *p* = 0.026) and all automated SIDCA plans (*p* = 0.012 or lower). In comparison to the separate-isocenter plans, the GI of the IMRT (*p* = 0.008) and SIDCA_1 plans with three angles (*p* = 0.013) were significantly higher. Moreover, the GI of the IMRT plans was significantly higher than the SIDCA_3 with high (3.4 ± 0.4, *p* = 0.001) and SIDCA_4 with low (3.8 ± 0.6, *p* = 0.013) sparing of the uninvolved brain. The GI of SIDCA_2 plans with five angles and the SIDCA_3 with high and SIDCA_4 with low sparing of the uninvolved brain did not differ significantly from the separate-isocenter plans (paired T-test). For ten patients irradiated in a single fraction, the GI was comparable with the results for 12 patients.

The mean dose to the uninvolved brain for the SIDCA_2 with five angles (3.0 ± 2.0 Gy) was comparable to the manual separate-isocenter plans (3.1 ± 1.7 Gy). Based on the ANOVA, the mean dose of the uninvolved brain was not significantly different between different plans, neither for ten patients irradiated in a single fraction. However, based on paired T-tests, a statistically significant difference was found between the separate-isocenter DCA and VMAT plans (*p* = 0.038), between VMAT and automated SIDCA_2 plans with five couch angles (*p* = 0.044).

The total V_12Gy_ of the uninvolved brain, automatically generated SIDCA plans (50 ± 51 cm^3^ and 55 ± 54 cm^3^ for SIDCA_3 and SIDCA_4, respectively), were comparable with the manual DCA plans (54 ± 58 cm^3^). According to the ANOVA results, the total V_12Gy_ did not differ significantly across all planning techniques for either all patients or the ten single-fraction patients.

For the total V_5Gy_, the automatically generated SIDCA plans were comparable with the manual separate-isocenter plans. According to the ANOVA, the total V_5Gy_ of the uninvolved brain was not significantly different between different plans, neither for ten patients irradiated in a single fraction. However, according to paired T-tests, a statistically significant difference was found between the separate-isocenter DCA and VMAT plans (*p* = 0.045), between VMAT and SIDCA_2 (*p* = 0.041) as well as SIDCA_3 with high NT sparing (*p* = 0.032).

The number of MUs was the highest for the separate-isocenter plans and significantly higher than for IMRT plans (*p* = 0.003) and other plans (*p* = 0.001 or lower). The number of MUs was the smallest for VMAT plans, followed by automatically generated SIDCA and IMRT plans and much lower than for the separate-isocenter DCA plans. According to the ANOVA, the total number of MUs is not significantly different between all SIDCA plans, neither for ten patients irradiated in a single fraction.

Irradiation time was longest for manually planned separate-isocenter DCA plans and significantly longer than for the IMRT plans (*p* = 0.035). According to the ANOVA, the irradiation time was not significantly different between the VMAT and all automated SIDCA plans (*p* = 0.12). The irradiation times of IMRT plans were significantly longer than those of the last five plans. Similar results were found for ten patients irradiated in a single fraction.

The pass rate was significantly lower for VMAT plans (87.7 ± 4.8) compared to separate-isocenter DCA (97.0 ± 2.5, *p* = 0.000), IMRT (99.7 ± 0.2, *p* = 0.000), SIDCA_1, SIDCA_2, and SIDCA_4 plans (*p* = 0.000). Also, according to the ANOVA, the pass rate was significantly different between the VMAT and all other plans (*p* = 0.000).

Boxplots of the maximum dose to brain stem, chiasm, optic nerves, and eyes are presented in Figure 4 for the seven planning techniques (see also Table 3). According to the ANOVA, the maximum dose to various OARs was not significantly different between different planning techniques. For the right eye, the maximum dose was statistically significantly different between different planning techniques (ANOVA 0.027 for 12 patients and 0.044 for ten patients). According to paired T-test, the maximum dose to the right eye of 2.7 Gy was higher for IMRT plans than for VMAT plans (*p* = 0.043) and for SIDCA_1, SIDCA_2, and SIDCA_4 plans than for separate-isocenter DCA plans (*p* = 0.015–0.029), but this value is still very low. For all patients and all plans, the maximum doses to OARs were within our limits and thus clinically acceptable, but varied between treatment techniques (see Figure 4 and Appendix A).

## 4. Discussion

Recent international guidelines proposed SRS also in patients with more than three BM with low-volume disease. Optimal treatment quality with sparing of healthy brain tissue is essential to avoid SRS/SRT complications such as brain necrosis. We found that automated planning with DCA or IMRT can make linac-based SRS/SRT plan quality with single isocenter comparable with a manually planned DCA plan with a separate isocenter for each metastasis. The Paddick GI was significantly worse for VMAT plans, because island blocking (open leaves between metastasis) is not automatically suppressed in Pinnacle (see an example of BEV in Figure 5a). Dynamic IMRT prevents island blocking between metastasis by automatic choice of collimator angles in iPlan (see Figure 5b). To prevent unnecessary irradiation of NT and decrease the dosimetric dependency between PTVs, MBM Elements uses two rules following from the definition of the blocking: (a) PTVs packed to an arc may not geometrically overlap for a control point; (b) PTVs packed to an arc may never use the same leaf pair for a control point as the gap would introduce unnecessary NT dose (see in Figure 5c). The benefit of automated SIDCA planning techniques was mainly observed in the lower dose region (V_5Gy_); in the V_12Gy_, no statistically significant differences were found. Moreover, automatically generated linac-based SRS/SRT plans result in a lower number of MUs and are more time-efficient compared to manual separate-isocenter DCA plans. The pass rate in Mobius was significantly lower for VMAT plans compared to separate-isocenter DCA, IMRT, and SIDCA plans (except SIDCA_3).

Previously published studies [13,14,15,17,19] compared automated planning with VMAT or separate-isocenter DCA, whereas we additionally performed a comparison of four types of automated SIDCA plans with dynamic IMRT. Most of these plans were calculated for a Varian or Novalis accelerator, whereas only a few comparisons were performed for an Elekta accelerator [13,15]. For linac-based SRS of multiple BM, all automated single-isocenter DCA and IMRT plans had lower peripheral dose spread (lower GI and lower V_5Gy_) than VMAT plans with comparable CI in our study. Gevaert et al. [15] drew the same conclusion about the peripheral dose of single-isocenter DCA in comparison to VMAT, using Eclipse RapidArc (Varian). Ruggieri et al. [33] concluded that HyperArc (Varian) resulted in a higher Paddick CI and a lower GI than standard multiple-isocenter VMAT plans. In our study, the Paddick GI was significantly worse for VMAT plans because island blocking is not automatically suppressed in Pinnacle.

According to Narayanasamy et al., multiple BM planning software produced equivalent conformity, dose falloff, and brain V_12Gy_, but required a significantly lower number of MUs when compared to RapidArc plans [14]. We found that SIDCA plans resulted in a lower number of MUs and irradiation time with an even better plan quality compared to manual DCA plans with the separate-isocenter approach.

Moreover, in our study, seven out of twelve patients had a total PTV > 15 cm^3^ and two patents were treated with three fractions, whereas the previously published studies according to Yamamoto et al. [3] included patients with smaller metastases treated with a single fraction. In contrast to the results of Vergalasova et al. [34], we did not find that SIDCA had worse CI in comparison to VMAT. In their study, for an easier comparison with HyperArc plans, the clinically used SIDCA plans were renormalized to the value used for HyperArc. In this way, the GI and CI values are influenced and will lead to worse results compared to the initial values derived from the original clinical plans. Moreover, Vergalasova et al. [34] studied smaller metastases (only one patient with the total PTV > 15 cm^3^). Hofmaier et al. [19] also compared automated SIDCA with single-isocenter VMAT (Monaco^®^, Elekta, Stockholm, Sweden) for smaller metastasis (≤11.9 cm^3^). They found that MBM Elements can often generate treatment plans with steeper dose gradients, superior healthy brain sparing and fewer MUs as compared to VMAT plans and recommend to perform such study with larger and more irregularly shaped lesions. This is exactly what we did in our study and found that automated SIDCA results in plans with significantly better GI and shorter irradiation time, not only for smaller metastases but also for large metastases. Also, we studied the V_12Gy_/(Total PTV) and V_5Gy_/(Total PTV) of the uninvolved brain, and the gradient index as a function of the total PTV (see Figure 6). The smallest total PTVs demonstrate the highest spread. For PTV > 8.6 cm^3^, a plateau for V_5Gy_/(Total PTV) and gradient index is seen in Figure 6b,c However, the volume irradiated with 12 Gy, V_12Gy_, shows a tendency to increase more than if there would be a linear relation with the PTV, as shown in Figure 6a. In any case, VMAT plans show the highest values for all PTVs. SIDCA_3 (high sparing of normal tissue) and SIDCA_4 (low sparing of normal tissue) plans with 5 couch angles show the lowest values for all PTVs.

One would expect a better OARs sparing with separate-isocenter DCA plans because of a possibility to avoid the critical structures. According to ANOVA in our study, the maximum dose to brain stem, chiasm, optic nerves, and eyes were not statistically significantly different between the seven planning techniques. This result was not published before.

Our study has some limitations. A small number of patients was analyzed. Additionally, it is probably possible to get better results with VMAT than SIDCA using another treatment planning system.

## 5. Conclusions

In conclusion, automatically generated linac-based single-isocenter SRS/SRT plans for multiple BM reduce the number of MUs and irradiation time with a comparable GI relative to the separate-isocenter DCA plans, which were our reference. SIDCA with 5 couch angles results in the best plan quality (lower GI and V_5Gy_) compared to VMAT and IMRT. According to ANOVA, the maximum dose to brain stem, chiasm, optic nerves, and eyes were not statistically significantly different between the seven planning techniques.

## Figures and Tables

**Figure 1 cancers-13-03458-f001:**
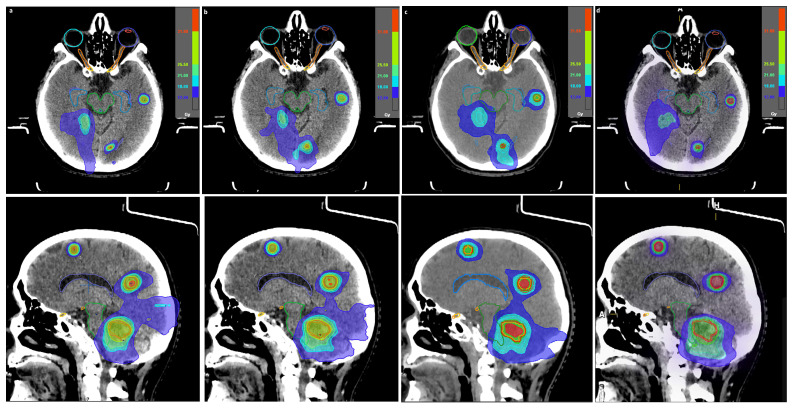
Dose distribution in axial (top) and sagittal (bottom) plane of patient 9 for separate-isocenter dynamic conformal arc (DCA) (**a**), intensity modulated radiotherapy (IMRT) (**b**), volumetric modulated arc radiotherapy (VMAT) (**c**), and Multiple Brain Mets SRS Elements (MBM Elements) v.2.0 with high normal tissue (NT) (**d**) plans. From the four MBM Elements plans, only the plan with the lowest gradient index is presented as representative from those plans.

**Figure 2 cancers-13-03458-f002:**
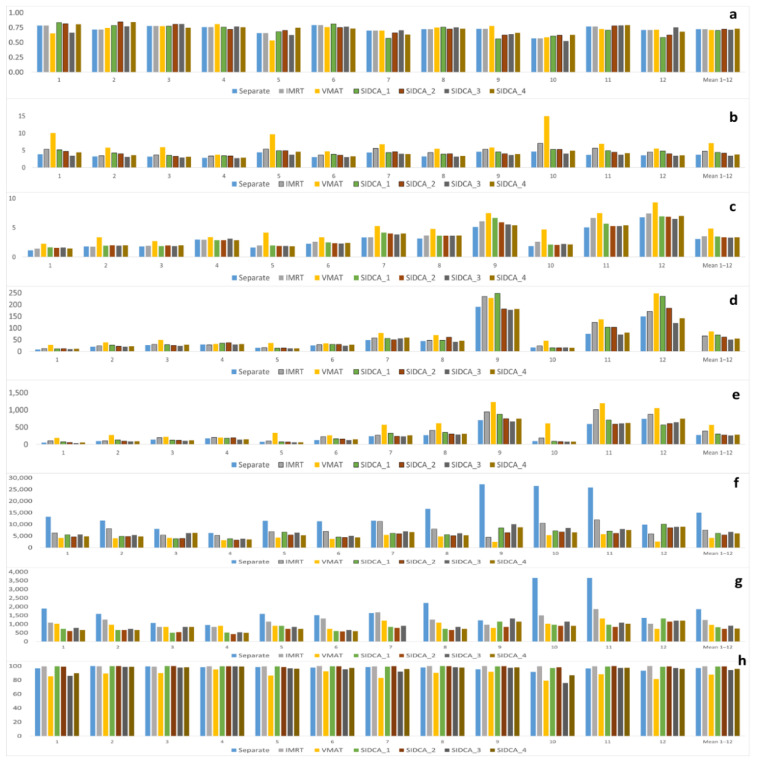
Comparison of the crucial parameters for different patients and planning techniques. Diagrams of the CI (**a**), the GI (**b**), the mean dose (**c**), the total V_12Gy_ (**d**) and V_5Gy_ (**e**) of the uninvolved brain, the number of MUs (**f**), irradiation time (**g**), and pass rate (3%/3 mm) (**h**) for separate-isocenter DCA (separate), IMRT, VMAT, MBM Elements v.1.6 with three couch angles (SIDCA_1), MBM Elements v.1.6 with five couch angles (SIDCA_2), MBM Elements v.2.0 with high NT (SIDCA_3), and low NT (SIDCA_4) plans for 12 patients and the mean values over these patients.

**Figure 3 cancers-13-03458-f003:**
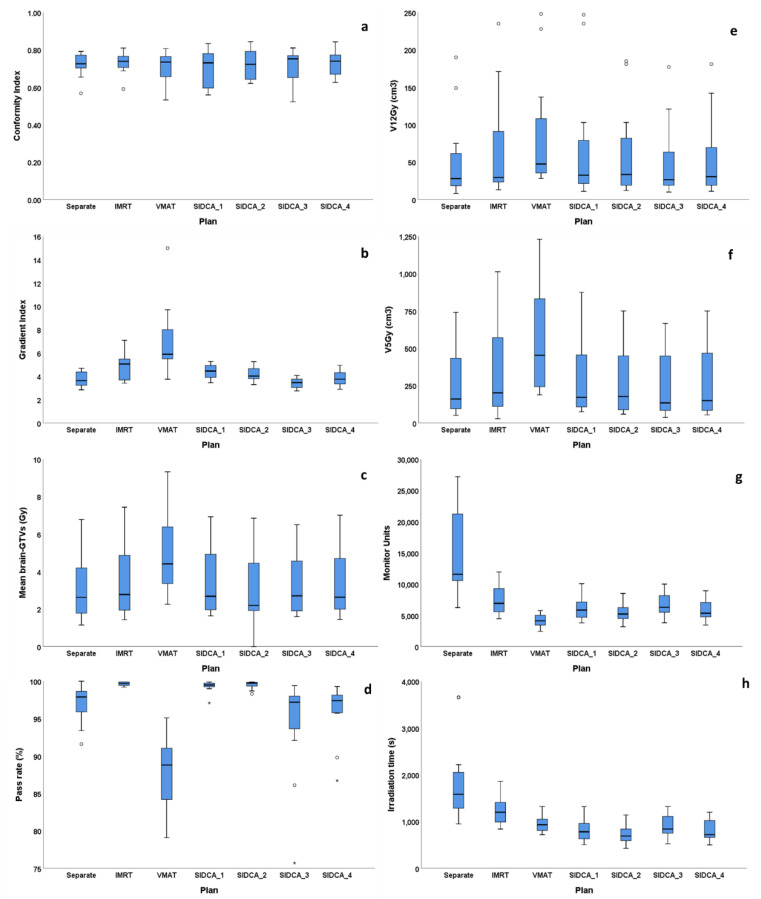
Statistical comparison of the crucial parameters for different planning techniques based on twelve patients. Box plots of the CI (**a**), the GI (**b**), the mean dose (**c**), pass rate (3%/3 mm) (**d**), the total V_12Gy_ (**e**) and V_5Gy_ (**f**) of the uninvolved brain, the number of MUs (**g**), and irradiation time (**h**) as a function of different plans. The central line of each box represents the median value, its upper and lower edges the 25th and 75th percentiles. The T-bars are called whiskers and extend to 1.5 times the height of the box values per planning technique. Outliers (circles) are defined as values that do not fall in the whiskers. Extreme outliers (more than 3 box lengths from either hinge) are marked with an asterisk (*) on the boxplot.

**Figure 4 cancers-13-03458-f004:**
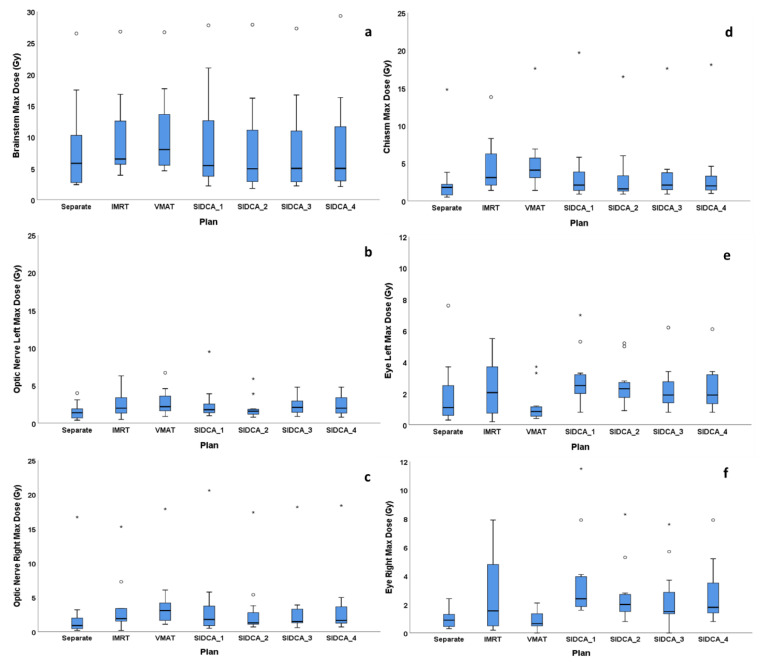
The boxplots of the maximum dose to brain stem (**a**), optic nerves (left **b**, right **c**), chiasm (**d**), and eyes (left (**e**), right (**f**)) as a function of different plans. * For all patients and all plans, the maximum doses to OARs were within our limits and thus clinically acceptable, but varied between treatment techniques.

**Figure 5 cancers-13-03458-f005:**
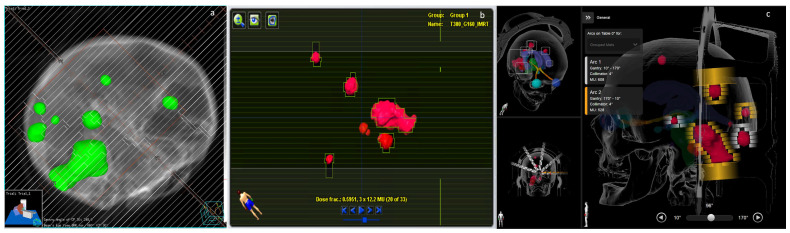
The beam’s eye view of VMAT arc (**a**), dynamic IMRT field (**b**), and MBM Elements v.2.0 arc (**c**).

**Figure 6 cancers-13-03458-f006:**
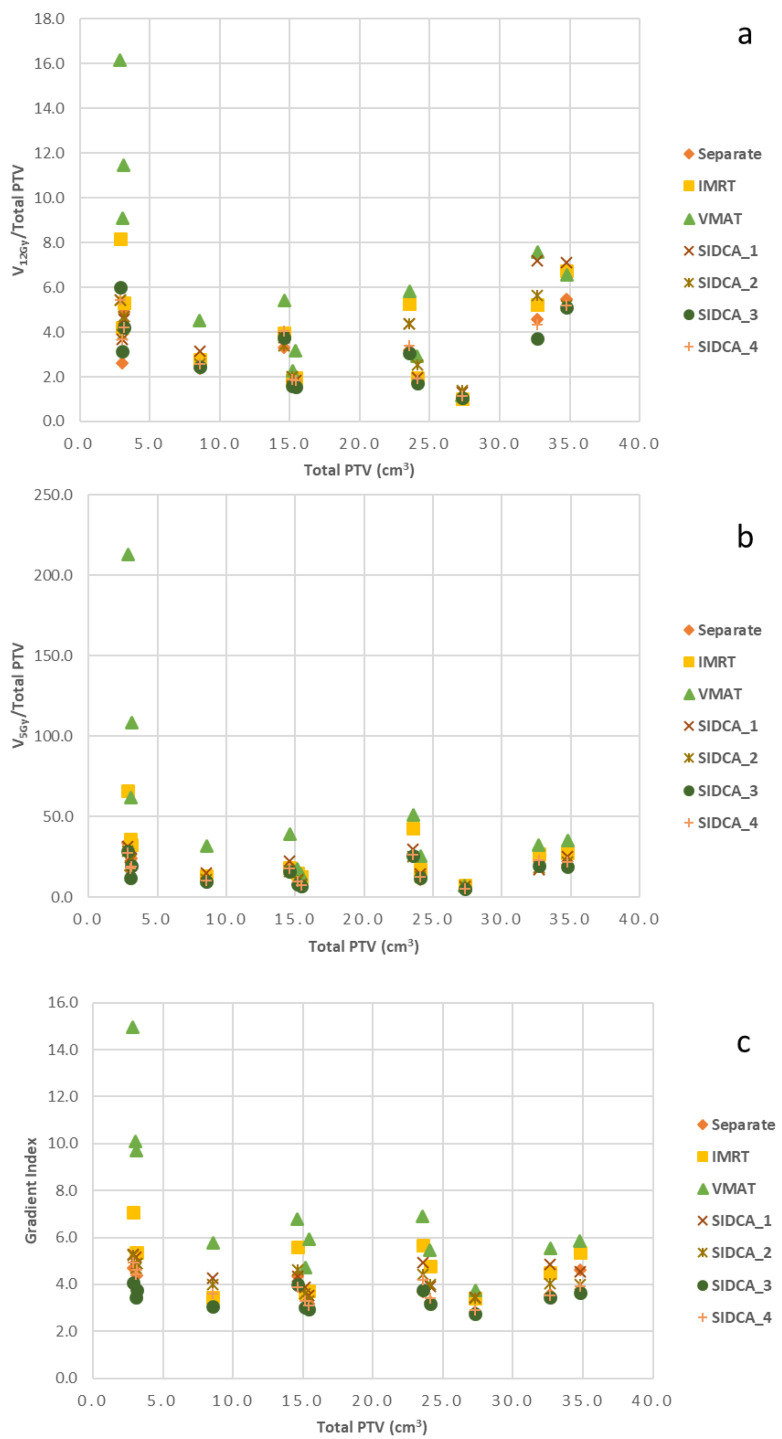
Graphs of the V_12Gy_/(Total PTV) (**a**) and V_5Gy_/(Total PTV) (**b**) of the uninvolved brain, and the gradient index (**c**) as a function of the total PTV for different plans.

**Table 1 cancers-13-03458-t001:** The number of metastases, the total planning target volume (PTV), all PTVs, and the prescription dose for the 12 studied patients.

Patient	Prescription	Total PTV (cm^3^)	PTV for Each Metastasis (cm^3^)									
			1	2	3	4	5	6	7	8	9	10
1	1 × 21 Gy	3.05	**1.16**	0.14	1.6	0.15	-	-	-	-	-	-
2	1 × 21 Gy	8.56	2.61	1.84	3.53	0.58	-	-	-	-	-	-
3	1 × 18 Gy	**15.42**	**13.5**	0.49	0.34	1.09	-	-	-	-	-	-
4	1 × 15 Gy	**27.30**	0.25	**21.14**	2.92	2.99	-	-	-	-	-	-
5	1 × 21 Gy	3.12	0.37	**1.04**	0.90	0.62	0.20	-	-	-	-	-
6	1 × 18 Gy	**15.17**	0.12	0.06	**11.72**	2.86	0.42	-	-	-	-	-
7	1 × 21 Gy	14.60	**6.41**	1.35	0.73	0.95	2.26	2.36	0.54	-	-	-
8	1 × 18 Gy	**24.09**	0.49	0.47	**10.68**	1.63	5.94	0.28	4.61	-	-	-
9	3 × 8.5 Gy	**34.78**	2.46	0.11	0.70	0.39	4.14	0.84	2.50	**23.64**	-	-
10	1 × 21 Gy	2.86	0.17	0.07	0.12	0.30	0.06	0.08	**1.60**	0.14	0.32	-
11	1 × 21 Gy	**23.55**	2.14	3.64	3.59	0.91	1.79	**7.29**	0.51	3.07	0.10	0.51
12	3 × 8.5 Gy	**32.68**	**15.60**	14.30	0.27	0.17	1.1	0.37	0.13	0.45	0.16	0.13

The total PTV per patient that are larger than 15 cm^3^ and the largest PTV are shown in bold. PTV, planning target volume.

**Table 2 cancers-13-03458-t002:** Conformity Index (CI), Gradient Index (GI), mean dose, total V_12Gy_ and V_5Gy_ of the uninvolved brain, i.e., brain volume—GTVs, number of monitor units (MUs), irradiation time and pass rate (3%/3 mm) as mean ± 1SD. The planning techniques, which were significantly different from separate-isocenter (reference) planning depending on the *p*-value, are shown with one, two, or three stars, corresponding to levels equal to or below 0.05, 0.01, and 0.001, respectively.

Plan	Conformity Index	Gradient Index	Mean Brain-GTVs (Gy)	V12Gy (cm^3^)	V5Gy (cm^3^)	MUs	Irradiation Time (s)	Pass Rate (%)
Separate-isocenter DCA many couch angles	0.72 ± 0.06	3.7 ± 0.6	3.1 ± 1.7	54 ± 58	276 ± 254	14,947 ± 907	1859 ± 7414	97.0 ± 2.5
IMRT 1 isoc 3 angles	0.73 ± 0.06	4.8 ± 1.1 **	3.5 ± 2.0	66 ± 71	388 ± 347	7517 ± 322 **	1235 ± 2468 **	99.7 ± 0.2 *
VMAT 1 isoc 3 angles	0.71 ± 0.08	7.1 ± 3.0 ***	4.9 ± 2.0 *	85 ± 77	563 ± 394 *	4123 ± 185 ***	955 ± 1066 **	87.7 ± 4.8 ***
SIDCA_1: v.1.6 MBM Elements 3 angles	0.70 ± 0.10	4.4 ± 0.6 *	3.5 ± 1.9	71 ± 83	305 ± 271	6130 ± 250 ***	820 ± 1881 ***	99.3 ± 0.8
SIDCA_2: v.1.6 MBM Elements 5 angles	0.72 ± 0.08	4.2 ± 0.6	3.0 ± 2.0	61 ± 62	273 ± 241	5403 ± 191 ***	724 ± 1434 ***	99.5 ± 0.5 *
SIDCA_3: v.2.0 MBM Elements high NT 5 angles	0.71 ± 0.08	3.4 ± 0.4	3.3 ± 1.7	50 ± 51	257 ± 242	6726 ± 236 ***	904 ± 1791 **	94.3 ± 6.9
SIDCA_4: v.2.0 MBM Elements low NT 5 angles	0.73 ± 0.07	3.8 ± 0.6	3.3 ± 1.8	55 ± 54	283 ± 268	5958 ± 383 ***	676 ± 1726 **	96.0 ± 3.8

* sign at ≤ 0.05, ** sign at ≤ 0.01, *** sign at ≤ 0.001.

**Table 3 cancers-13-03458-t003:** The maximum dose to brain stem, chiasm, optic nerves (left and right), and eyes (left and right) averaged over twelve patients as mean ± 1SD. For all planning techniques, they were not significantly different from separate-isocenter (reference) planning (*p* > 0.05).

Plan	Maximum Dose (Gy)						
	Brain Stem	Chiasm	Optic Nerve Left	Optic Nerve Right	Eye Left	Eye Right	Pituitary Gland
Separate-isocenter DCA many couch angles	8.2 ± 7.8	2.8 ± 4.1	1.6 ± 1.1	2.5 ± 4.8	2.1 ± 2.2	1.0 ± 0.7	2.5 ± 3.5
IMRT 1 isoc 3 angles	9.8 ± 7.1	4.6 ± 3.9	2.6 ± 1.9	3.6 ± 4.4	2.3 ± 1.9	2.7 ± 2.8	3.7 ± 3.3
VMAT 1 isoc 3 angles	10.5 ± 7.1	5.2 ± 4.4	2.8 ± 1.7	4.3 ± 4.7	1.2 ± 1.1	0.9 ± 0.6	3.9 ± 3.1
SIDCA_1: v.1.6 MBM Elements 3 angles	9.4 ± 8.6	4.0 ± 5.4	2.6 ± 2.4	3.8 ± 5.9	2.9 ± 1.8	3.8 ± 3.1 *	4.2 ± 4.8
SIDCA_2: v.1.6 MBM Elements 5 angles	8.4 ± 8.1	3.5 ± 4.6	2.0 ± 1.5	3.3 ± 4.9	2.6 ± 1.4	2.7 ± 2.2 *	3.3 ± 3.5
SIDCA_3: v.2.0 MBM Elements high NT 5 angles	8.0 ± 7.7	3.6 ± 4.5	2.3 ± 1.3	3.3 ± 4.8	2.3 ± 1.5	2.5 ± 2.3	3.1 ± 3.1
SIDCA_4: v.2.0 MBM Elements low NT 5 angles	8.3 ± 8.2	3.5 ± 4.7	2.4 ± 1.4	3.4 ± 4.9	2.3 ± 1.5	2.8 ± 2.2 *	3.1 ± 3.2

* sign at ≤ 0.05.

## Data Availability

Original data are available by request from the corresponding author.

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
