# Peer review of "Quality of Automated Stereotactic Radiosurgery Plans in Patients with 4 to 10 Brain Metastases"

_cancers, 2021, doi:10.3390/cancers13143458_

Round 1

Reviewer 1 Report

In this article entitled “Quality of Automated Stereotactic Radiosurgery Plans in Patients with 4 up to 10 Brain Metastases”, the Authors investigated whether linac-based SRS/SRT plan quality of single-isocenter plans, either automatically generated or using dynamic IMRT, could be comparable or superior to the plan quality of manually planned DCA-plans with separate isocenters for each metastasis as a reference.

The AAs evaluated 12 patients with 4 up to 10 brain metastases (< 4 cm in diameter), with a Karnofsky performance score > 60 and with a prognosis of more than six months. A total of 77 lesions were selected for this study.

For many years whole-brain radiotherapy has been the standard treatment of multiple brain metastases, however this technique has many side effects and a low local tumor control probability. SRS is a promising treatment option for patients with four or more brain metastases and recent international guidelines proposed SRS also in patients with more brain metastases with low volume disease.

In the literature data, the SRT may decrease late side effects compared to SRS

Using seven planning techniques, the AAS founded that automated planning with DCA or IMRT can make linac-based SRS/SRT plan quality with a single isocenter comparable with a manually planned DCA plan with a separate isocenter for each metastasis.

In the current era of personalized medicine, the AAs explored a major interest subject about treatment options for patients with multiple brain metastases. The article is interesting in its content but analyzed a small number of patients.

The abstract shows the purposes of the AAs and may be considered a good summary. Methods and results appear to be well exposed in terms of logic and organization and carefully described. The AAs attempted to explain and discuss the different data carefully. The conclusions summarize very well the results achieved and are well-written and concise.

Limitations of the study are missing.

The AAs refer to valid and appropriated references. The tables and figures adequately show the data explained in the text.

Author Response

Thank you very much for your positive evaluation and comments. In this document we duplicated the reviewer’s comments and added our responses to each of them (in blue). Furthermore, in the revised manuscript we highlighted the sections that we changed / added. Comments and Suggestions for Authors Reviewer 1 In this article entitled “Quality of Automated Stereotactic Radiosurgery Plans in Patients with 4 up to 10 Brain Metastases”, the Authors investigated whether linac-based SRS/SRT plan quality of single-isocenter plans, either automatically generated or using dynamic IMRT, could be comparable or superior to the plan quality of manually planned DCA-plans with separate isocenters for each metastasis as a reference. The AAs evaluated 12 patients with 4 up to 10 brain metastases (< 4 cm in diameter), with a Karnofsky performance score > 60 and with a prognosis of more than six months. A total of 77 lesions were selected for this study. For many years whole-brain radiotherapy has been the standard treatment of multiple brain metastases, however this technique has many side effects and a low local tumor control probability. SRS is a promising treatment option for patients with four or more brain metastases and recent international guidelines proposed SRS also in patients with more brain metastases with low volume disease. In the literature data, the SRT may decrease late side effects compared to SRS Using seven planning techniques, the AAS founded that automated planning with DCA or IMRT can make linac-based SRS/SRT plan quality with a single isocenter comparable with a manually planned DCA plan with a separate isocenter for each metastasis. In the current era of personalized medicine, the AAs explored a major interest subject about treatment options for patients with multiple brain metastases. The article is interesting in its content but analyzed a small number of patients. We would like to thank the reviewer for the positive reaction about the topic of our manuscript. A small number of patients was indeed analysed and we have added this point as a limitation of the manuscript. In addition, we believe that our results are sufficiently interesting and novel in comparison with the previously published reports. The abstract shows the purposes of the AAs and may be considered a good summary. Methods and results appear to be well exposed in terms of logic and organization and carefully described. The AAs attempted to explain and discuss the different data carefully. The conclusions summarize very well the results achieved and are well-written and concise. Limitations of the study are missing. We would like to thank the reviewer for the recommendation to add limitations and completely agree with this suggestion. Limitations have been added to the Discussion section of the manuscript. The AAs refer to valid and appropriated references. The tables and figures adequately show the data explained in the text.

Reviewer 2 Report

The article is well written, data are shown clearly and results are conform to the authors purpose. results are in line with the literature. Introduction of an additional isocenter appears to partly mitigate severe target underdosage: it is reported especially for smaller target sizes such as in >4 mts. Patients follow up should be enclosed in the paper to better enstablish clinical effect of radiation necrosis.

Author Response

Thank you very much for your positive evaluation and comments. In this document we duplicated the reviewer’s comments and added our responses to each of them (in blue). Furthermore, in the revised manuscript we highlighted the sections that we changed / added. Comments and Suggestions for Authors Reviewer 2 The article is well written, data are shown clearly and results are conform to the authors purpose. results are in line with the literature. Introduction of an additional isocenter appears to partly mitigate severe target underdosage: it is reported especially for smaller target sizes such as in >4 mts. Patients follow up should be enclosed in the paper to better enstablish clinical effect of radiation necrosis. We would like to thank the reviewer for their positive evaluation and for the recommendation to add patients follow up and completely agree with this suggestion but it was not a part of this planning study. We are planning another study comparing the follow up of a larger group of patients with multiple brain metastases treated in a single and multiple isocenters.
